# Transformer-Based Decoder Designs for Semantic Segmentation on Remotely Sensed Images

**Teerapong Panboonyuen** [1], **Kulsawasd Jitkajornwanich** [2], **Siam Lawawirojwong** [3], **Panu Srestasathiern** [3] and **Peerapon Vateekul** [1,*]

1 Department of Computer Engineering, Faculty of Engineering, Chulalongkorn University, Phayathai Rd, Pathumwan, Bangkok 10330, Thailand; teerapong.panboonyuen@gmail.com
2 Data Science and Computational Intelligence (DSCI) Laboratory, Department of Computer Science, King Mongkut's Institute of Technology Ladkrabang, Chalongkrung Rd, Ladkrabang, Bangkok 10520, Thailand; kulsawasd.ji@kmitl.ac.th
3 Geo-Informatics and Space Technology Development Agency (Public Organization), 120, The Government Complex, Chaeng Wattana Rd, Lak Si, Bangkok 10210, Thailand; siam@gistda.or.th (S.L.); panu@gistda.or.th (P.S.)
* Correspondence: peerapon.v@chula.ac.th

**Abstract:** Transformers have demonstrated remarkable accomplishments in several natural language processing (NLP) tasks as well as image processing tasks. Herein, we present a deep-learning (DL) model that is capable of improving the semantic segmentation network in two ways. First, utilizing the pre-training Swin Transformer (SwinTF) under Vision Transformer (ViT) as a backbone, the model weights downstream tasks by joining task layers upon the pretrained encoder. Secondly, decoder designs are applied to our DL network with three decoder designs, U-Net, pyramid scene parsing (PSP) network, and feature pyramid network (FPN), to perform pixel-level segmentation. The results are compared with other image labeling state of the art (SOTA) methods, such as global convolutional network (GCN) and ViT. Extensive experiments show that our Swin Transformer (SwinTF) with decoder designs reached a new state of the art on the Thailand Isan Landsat-8 corpus (89.8% *F*1 score), Thailand North Landsat-8 corpus (63.12% *F*1 score), and competitive results on ISPRS Vaihingen. Moreover, both our best-proposed methods (SwinTF-PSP and SwinTF-FPN) even outperformed SwinTF with supervised pre-training ViT on the ImageNet-1K in the Thailand, Landsat-8, and ISPRS Vaihingen corpora.

**Keywords:** vision transformer; fully transformer networks; convolutional neural network; feature pyramid network; high-resolution representations; ISPRS Vaihingen; Landsat-8

## 1. Introduction

In general, automated semantic segmentation is studied to analyze remote sensing [1–3]. Research into semantic segmentation of aerial or satellite data has grown in importance. Over the years, due to its full range of autonomous driving, automatic mapping, and navigation application, significant progress has been made in this field. In the last decade, DL has been revolutionized by computer science. Among modern convolutional neural networks (ConvNet/CNNs), there are many techniques, e.g., dual attention [4] and self-attention [5], that have gained increasing attention due to their capability. Such techniques generate highly precise semantic segmentation from remote sensing data. Still, all suffer from issues regarding the accuracy of performance.

Currently, many deep learning architectures [2,6] have been applied in urban or agriculture segmentations, such as global convolutional networks [7], DeepLab [8], mask R-CNN [9], BiseNet [10], and CCNet [11]. These networks have been created for semantic recognition and consist of stacked convolution blocks. Due to reduced costs of computation, the use of kernel maps has decreased gradually.

Thus, the encoder network can learn more semantic visual theories with a steadily increased receptive field. Consequently, this also inflates a primary restriction of studying long-range dependency knowledge, which is significant for computer vision tasks. However, the situation is still challenging due to the limited size of the region in the input that produces the feature. These receptive fields require dense high-resolution predictions; transformers conduct self-attention on that receptive field. Previously, architecture has not fully leveraged various feature maps from convolution or attention blocks conducive to image segmentation, and this was a motivation for this work.

To overcome this weakness, completely new networks viz. Swin Transformer (SwinTF) [12] with Vision Transformer (ViT) [13] as the major backbone, have a tremendous capacity in long-range dependency acquisition and sequence-based picture modeling. Transformers are the first transduction models that rely entirely on self-attention to compute their input and output representations without using sequence-aligned RNNs or convolution. No recurrent units are used to obtain these features; they are simply weighted sums and activations, which prove to be very efficient and parallelizable [14].

ViT is one of the most well-known Transformers used in several computer vision tasks, such as hyperspectral image classification [15,16], bounding-box detection [17,18], and semantic segmentation [19,20]. ViT moves the window divider between successive levels of self-attention. The shifted windows provide links between the windows of the last layer, considerably increasing modeling capability.

Most relevant to our proposed method is the Vision Transformer (ViT) [13] and their follow-ups [21–25]. ViT is a deep learning architecture that utilizes the mechanism of attention, focusing on image recognition and is greatly valued in their works [21–25]. Several works of ViT directly employ a transformer model on non-overlapping medium-sized image patches for image classification. ViT reaches an exciting speed-performance trade-off on almost all computer vision tasks compared to previous DL networks. DeiT [26] introduces several training policies that also allow it to be efficient using the extra modest ImageNet-1K corpus.

The effects of ViT on computer vision tasks are encouraging. The ViT model is inappropriate for low-resolution kernel filters and the image size's quadratic improvement in complexity. Some works utilize ViT models for the dense image tasks of semantic segmentation and detection. Notably, ViT [12,27] models are seen to have the best performance-accuracy trade-offs among these methods on computer vision tasks, even though this work concentrates mostly on general-purpose performance rather than focusing on semantic segmentation.

Moreover, it usually takes high computational costs for the previous transformer network, e.g., Pyramid ViT [28], which is quadratic to the size of an image. In contrast, SwinTF has solved the computational issue and costs linear to the image size. SwinTF has improved the accuracy by operating the model regionally, enhancing receptive fields that highly correlate to visual signals. Furthermore, it is efficient and effective, achieving SOTA performance, e.g., *MeanIoU*, *AveragePrecision* on COCO object detection, and ADE20K semantic segmentation.

In this paper, transformer-based decoder designs for multi-object segmentation from medium-resolution (Landsat-8) and very high-resolution (aerial) images are introduced, as demonstrated in Figures 1 and 2. This work helps to further improve SOTA on semantic segmentation in Landsat-8 and aerial images. For better performance, three styles of decoder designs into transformer-based reasoning are implemented. Our goals are two-fold:

- Utilizing a pre-training ViT to retrieve the virtual visual tokens based on the vision patches from aerial and satellite images: we immediately fine-tune the model weights on downstream responsibilities by appropriating pre-training SwinTF under ViT, as a backbone, by appending responsibility layers and superimposing the pretrained encoder.

- Proposing the decoder designs to our DL network with three decoder designs including (i) U-Net [29], (ii) pyramid scene parsing (PSP) network [30], and (iii) feature pyramid network (FPN) [31] to perform pixel-level segmentation.

The experimental results on three remotely sensed semantic segmentation corpora, including two Thailand Landsat-8 data sets and one ISPRS Vaihingen [32] corpora, demonstrate the effectiveness of the proposed scheme. The results prove that our SwinTF with decoder designs can overcome the previous encoder–decoder network [33–36] on aerial and satellite images and Swin Transformer models [12] in terms of the *Precision*, *Recall*, and *F1* score sequentially.

The remainder of this article is structured as follows. Section 2 discusses the materials and methods. The results are detailed in Section 3, and Section 4 presents our discussion, including our limitations and outlook. Finally, our conclusions are drawn in Section 5.

## 2. Material and Methods

### 2.1. Transformer Model

2.1.1. Transformer Based Semantic Segmentation

SwinTF follows a sequence-to-sequence vector with transformers [37] as well as a corresponding output vector with input vector fabrication, such as NLP. NLP concerns the interaction between computers and human language in order to process and analyze a large amount of matured language. Accordingly, the SwinTF, as described in Figure 1 allows a $1D$ sequence of vector embeddings $z \in R^{L \times C}$ as input, $L$ is the length of the vector, and $C$ is the hidden kernel size. The image sequence is consequently obliged to modify an input layer of image $x \in R^{H \times W \times 3}$ into $Z$.

The traditional SwinTF model [12] focuses on the relationship between a token (image patches); the other tokens are calculated. ViT focuses on the quadratic complexity concerning the number of image patches; finding it unsuitable for many image problems requiring an immense set of tokens for the softmax layer.

A traditional transformer-based encoder learns vector representations as to the $1D$ vector of embedding sequence $E$ input. This means that each ViT layer has a global receptive field, which answers the insufficient receptive field problems of the existing encoder–decoder deep neural network. The ViT encoder consists of $L_e$ layers of multilayer perceptron (MLP) and multi-head self-attention (MSA) modules.

A method for the sequence of image vectors is to flatten the pixel of values of images within a $1D$ vector with a size of $3 \times H \times W$. For a representative image, i.e., $512(H) \times 512(W) \times 3$, the resulting vector will have a length of 786,432. It is not conceivable that such high-dimensional vectors can be handled in both time and vector space. Accordingly, tokenizing every pixel of the image as input to our SwinTF is subject to a linear embedding layer.

In the case whereby a conventional encoder designed for semantic segmentation would downsample a $2D$ image $x \in R^{HW3}$ into a grid via a featuremap $x_f \in R^{\frac{H}{16} \times \frac{W}{16} \times C}$, we decided to set the transformer input sequence length $L$ as $\frac{H}{16} \times \frac{W}{16} = \frac{W}{256}$. This means that the output of the vector sequence of ViT can be clearly reshaped to the point kernel map $x_f$.

To recover the $\frac{HW}{256}$-long vector sequence of our input, we divide the image $x \in R^{H \times W \times 3}$ into a grid of as $\frac{H}{16} \times \frac{W}{16}$ patches. Thus, several ViT modules with modified self-attention calculation (SwinTF modules) are adapted on these image patch tokens. The ViT module maintains the number of patches $\frac{H}{4} \times \frac{W}{4}$ and then makes a series out of this grid. Each vectorized patch p is mapped into a latent C-dimensional embedding space using a linear projection function. $f : p \rightarrow e \in R^C$, for a patch $x$; we obtain a 1D series of vector embeddings. Therefore, we obtain a unique embedding $p_i$ for each position $i$ to encode the patch spatial information, which is then added to $e_i$ to generate the final sequence input $E = \{e_1 + p_1, e_2 + p_2, ..., e_L + p_L\}$. In this process, spatial data is kept, notwithstanding the order-less attention type of transformers.

A classical transformer-based encoder accepts feature representations when given the 1D embedding sequence $E$ as input. This encoder means that each ViT layer has a global

receptive field, resolving the problem of the standard deep learning encoder's restricted sensory area once and for all. The encoder of SwinTF consists of $L_e$ vector of MLP and MSA modules (Figure 1). At each layer $l$, the input to self-attention is depicted as a triplet of $(query, key, value)$, and calculated from the input $Z^{l-1} \in R^{L \times C}$ as:

$$query = Z^{l-1}W_Q, key = Z^{l-1}W_K, value = Z^{l-1}W_V \tag{1}$$

where $W_Q/W_K/W_V \in R^{C \times d}$ are the learnable weights of three linear projection vectors and $d$ is the dimension of $(query, key, value)$. Self-attention (SA) is then expressed as:

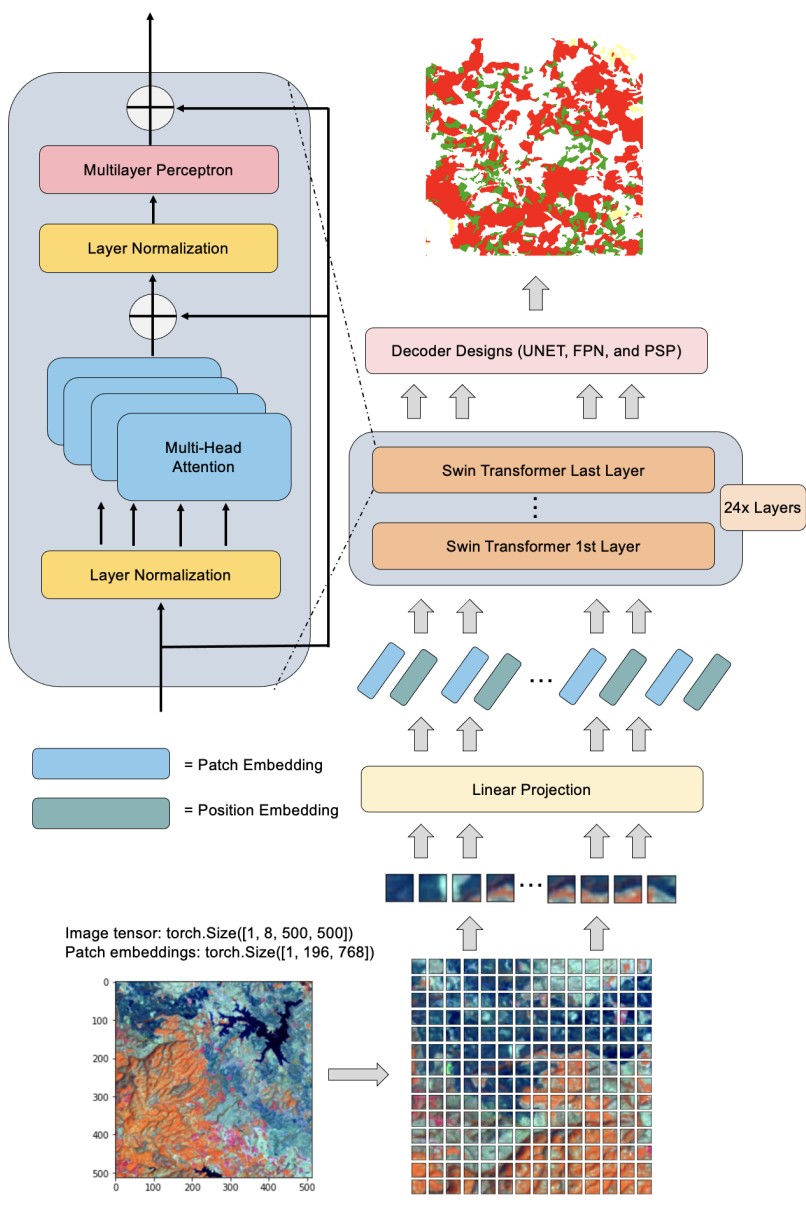

**Figure 1.** The overall architecture of our SwinTF.

$$SA(Z^{l-1}) = Z^{l-1} + softmax\left(\frac{Z^{l-1}W_Q(ZW_K)^T}{\sqrt{d}}\right)(Z^{l-1}W_V) \tag{2}$$

MSA clearly calculated a reckoning with m self-supporting SA actions and projects their concatenated outputs: $MSA(Z^{l-1}) = [SA_1(Z_l - 1); SA_2(Z_l - 1); \ldots; SA_m(Z_l - 1)]W_O$. Where $W_O \in R^{md} \times C$. $d$ is typically set to $C/m$. The output of MSA is then transformed by an MLP module with a residual skip as the output layer as:

$$Z^l = MSA(Z_{l-1}) + MLP(MSA(Z^{l-1})) \in R^{L \times C}. \tag{3}$$

Lastly, a normalized layer is employed before MLP and MSA modules, which are omitted for clearness. We express $Z^1, Z^2, Z^3, \ldots, Z^{L_e}$ as the weights of transformer vectors.

### 2.1.2. Decoder Designs

To assess the effectiveness of SwinTF's encoder vector, as represented by Z, three various decoder designs as portrayed in Figure 2 are set up to achieve pixel-level labeling. Next, the three decoders can be expressed as:

(1) U-Net [29]: The expansion route (decoder) on the right-hand side applies transposed convolutions with ordinary convolutions. The image size gradually increases in the decoder, whereas the depth gradually decreases. To improve precision, we employ the skip connections at every stage of the decoder by concatenating the output of the transposed convolution layers with the feature maps from the encoder at the same level. The encoder path's high-resolution (but semantically infirm) characteristics are mixed and reused with the upsampled output in this way.

As seen in the diagram below, U-Net has an asymmetrical design. Every step in the expanding direction, consisting of an upsampling of the feature map followed by a $2 \times 2$ transpose convolution that halves the number of feature channels, is used in the Decoder route. Accordingly, we have a concatenation with the contracting path's appropriate feature map, as well as a $3 \times 3$ convolutional neural network (each followed by a Rectified Linear Unit (ReLU)). A $1 \times 1$ convolution transfers the channels to the required number of classes in the final layer. Such a purpose is to bridge the feature gap between the decoder and encoder feature maps before concatenation.

(2) For pixel-level scene parsing, the PSP network is used and provides excellent global contextual prior [30]. The pyramid pooling module can capture more representative levels of data than global average pooling (GAP). The concept of sub-region average pooling is comparable to SPPNet's Spatial Pyramid Pooling [38]. Bilinear interpolation is employed to make all the feature maps' sizes equal; the 11 convolution then concatenation is akin to the depthwise convolution in Depthwise Separable Convolution utilized by Xception [39] or MobileNet [40]. To minimize the detrimental effect as much as possible, upsampling to $2\times$ is limited.

As a result, full-resolution from $BZ^{L_e}$ with size A$\frac{H}{16} \times \frac{W}{16}$ requires a total of four processes. The green layer, as seen in Figure 2, is the coarsest level, performing GAP over each feature map to provide a single bin output. The yellow layer is the second level, which divides the feature map into $2 \times 2$ sub-regions and performs average pooling for each of them. The third level, the light blue layer, separates the feature map into 33 sub-regions before serving average pooling for each sub-region. Finally, each low-dimension feature map is up-sampled to the same size as the original feature map (last blue layer), followed by a convolution layer to produce the final prediction map.

(3) FPN [31] is a characteristic extractor created with accuracy and speed in mind for such a pyramid idea. FPN takes the place of detectors, like Faster R-feature CNN's extractor [41]. Image recognition generates many feature map layers (multi-scale feature maps) and has superior quality to the traditional feature pyramid. FPN also utilizes specifically constructed transformers in a self-level, top-down, and bottom-up interactive pattern to change any feature pyramid into another feature pyramid of the same size but with richer contexts. The simple query, key, and value operation (Equation (1)) demonstrates its importance in choosing informative long-range interaction, which fits our objective of non-local interaction at appropriate sizes.

The higher-level feature using the visual qualities of the lower-level "pixels" is depicted. Each level's feature maps (red, yellow, and blue) are resized to their matching map size and concatenated with the original map before being sent to the convolution layer, which resizes them to the accurate "thickness". Higher-resolution features are upsampled from higher-pyramid-level feature maps, which are spatially coarser but semantically more robust. Spatial resolution is upsampled by a factor of two, with the nearest neighbor being used for simplicity. Each lateral link combines feature maps from the bottom-up and top-down paths of the same spatial size. To minimize the channel dimensions, the feature maps from the bottom-up course are convolutional (11 times).

In addition, element-wise addition is used to combine the feature maps from the bottom-up and top-down pathways. Finally, a 33 convolution is applied to each merged map to form the final feature map to reduce the aliasing impact of upsampling. This last collection of feature maps corresponds to the precise spatial dimensions. As all layers of the pyramid, as in a standard featured picture pyramid, employ joint classifiers/regressors, the feature dimension at output d is fixed at $d = 256$. As a result, the outputs of all further convolutional layers are 256-channel.

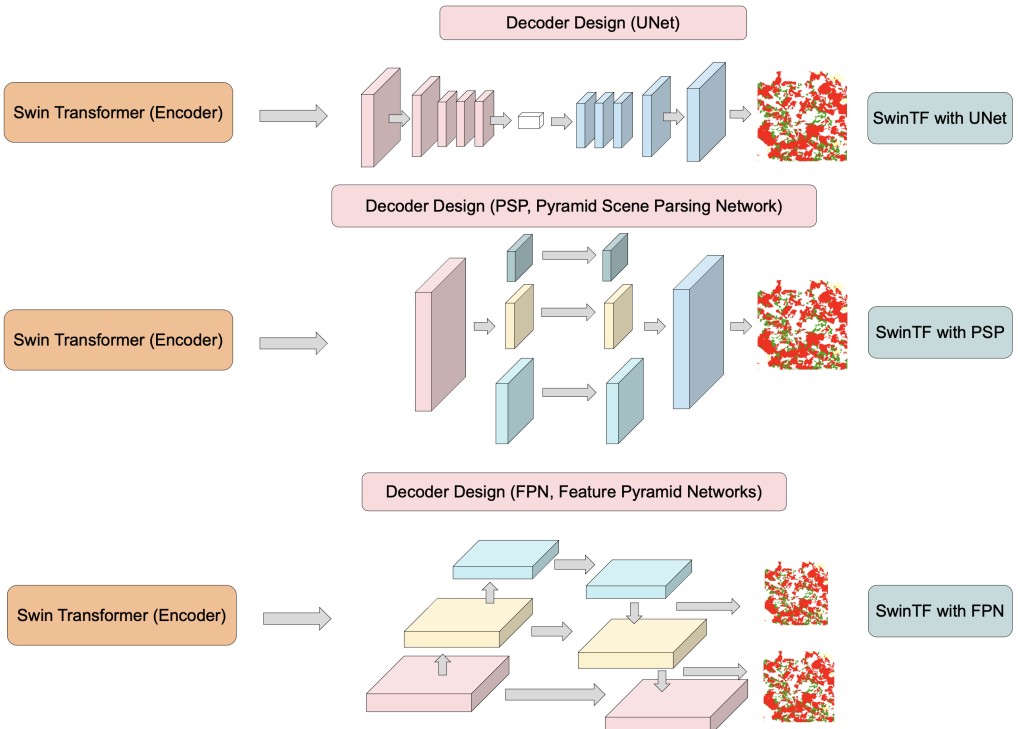

**Figure 2.** SwinTF with three variations of our decoder designs: SwinTF-UNet, SwinTF-PSP, and SwinTF-FPN.

2.1.3. Environment and Deep Learning Configurations

Herein, a stochastic depth dropout of 0.25 for the first 70% of training iterations is employed, and the dropout ratio to 0.6 is increased for the last 20%. As for the multi-scale flipping testing, testing scales of 0.5, 0.75, 1.0, 1.25, 1.5, and 1.75 are presented along with random horizontal flips by following standard practices, as in the literature (e.g., [12,13,31,37]) throughout training for all the experiments.

As the optimizer, a learning rate (LR) schedule is used with Stochastic gradient descent (often abbreviated SGD) for optimizing an the loss function with suitable smoothness properties. Weight decay and momentum are locked to 0.25 and 0.75, sequentially, for all the experiments on the three datasets. The initial LR of 0.0001 is set up on the Thailand Landsat-8 corpora and 0.001 on the ISPRS Vaihingen data set. Finally, batch normalization

in the fusion layers is employed and carried out using batch size 48. Images are resized to 512 pixels side length.

*2.2. Aerial and Satellite Imagery*

There are three primary sources of data in our experiments: one public and two private data sets. The private data sets are medium resolution imagery gathered from the satellite "Landsat-8" owned by the Thai government's Geo-Informatics and Space Technology Development Agency (GISTDA). As there are two different annotations, the Landsat-8 data is divided into two categories (Isan and North corpora), as illustrated in Table 1. The public data collection consists of high-resolution imagery from the "ISPRS Vaihingen (Stuttgart)" standard benchmark.

In our works, two types of data sets are used: satellite data and aerial data. Table 1 displays one aerial corpus (ISPRS Vaihingen data set) and two satellite data sets (TH-Isan Landsat-8 and TH-North Landsat-8 data sets). The Vaihingen data set contains 16 patches. Such data have been collected at particular locations with different sizes of resolution.

**Table 1.** Numbers of training, validation, and testing sets.

| Data Set | Total Images | Training Set | Validation Set | Testing Set |
| --- | --- | --- | --- | --- |
| TH-Isan Landsat-8 Corpus | 1420 | 1000 | 300 | 120 |
| TH-North Landsat-8 Corpus | 1600 | 1000 | 400 | 200 |
| ISPRS Vaihingen Corpus | 16 (Patches) | 10 | 2 | 4 |

2.2.1. North East (Isan) and North of Thailand Landsat-8 Corpora

The Isan district of Thailand's northeast is characterized by gently undulating topography, which mostly ranges in altitude from 90 to 180 m (300 to 600 feet), sloping from the Phetchabun Mountains in the west down to the Mekong River. The plateau is separated into different plains: the Mun and Chi rivers drain the southern Khorat plain, while the Loei and Songkhram rivers drain the northern Sakon Nakhon plain. The two tables are divided by the Phu Phan Mountains. The land is primarily sandy, with a great deal of salt deposits.

The north of Thailand is known for its varied landforms: low hills, crisscrossing mountains and valleys, with a large area of land suitable for cultivation, such as corn, pineapple, and para rubber. The Ping, Wang, Yom, and Nan rivers travel south through mountain valleys before uniting to form the Chao Phraya in Nakhon Sawan Province in central Thailand.

All the images in this data set were captured in Thailand's northern and Isan regions (Changwat). The Landsat-8 satellite contributed to the data gathering, which included 1420 and 1600 satellite images for the north data set and Isan data set, respectively, as seen in sampled data as in Figures 3 and 4. This data set has a massive collection of (46,128 × 47,616) pixel medium-quality images; corn (yellow), para rubber (red), and pineapple (green) are the three classes. A total of 1420 images are divided into 1000 training, 300 validation images, and 120 test images for the northern corpus. A total of 1600 images are divided into 1000 training, 400 validation images, and 200 test images for the Isan corpus for comparability with other baseline methods.

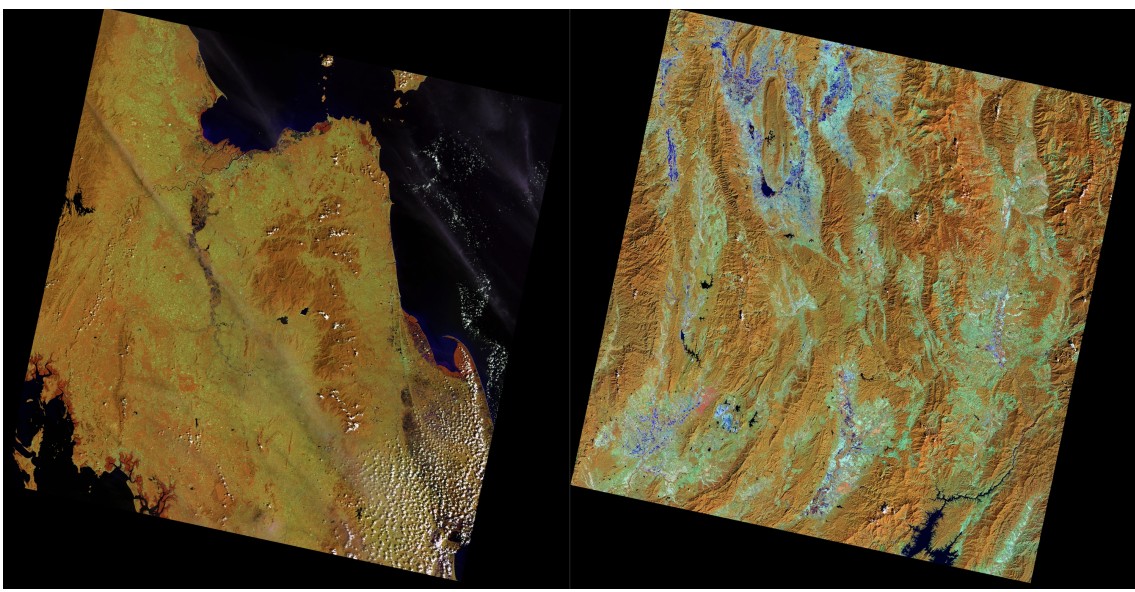

**Figure 3.** An illustration of a Landsat-8 scene (northern province (**left**) and northeastern region (**right**).

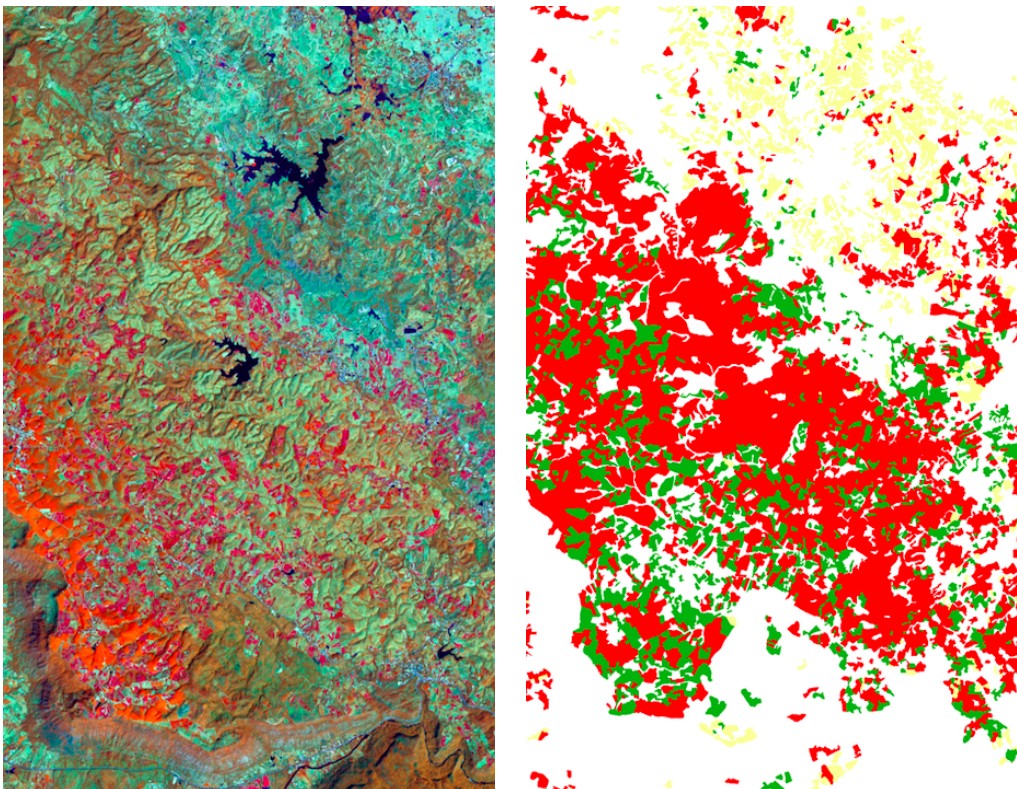

**Figure 4.** The left image is a sample of the northern province, and the right is the target image from the TH-Isan Landsat-8 corpus. Three classes comprise the target of the medium-resolution data set: para rubber (red), corn (yellow), and pineapple (green).

### 2.2.2. ISPRS Vaihingen Corpus

Our benchmark dataset is the ISPRS semantic segmentation challenge [32] (Figures 5 and 6) in Vaihingen (Stuttgart). They seized command of the German city of Vaihingen. The ISPRS Vaihingen corpus contains 3-band IRRG (Red, Infrared, and Green) image data, corresponding NDSM (Normalized Digital Surface Model), and DSM (Digital Surface Model) data. The latter highlights 33 scenes with a resolution of approximately 2500 × 2000 pixels and a capacity of

about 9 cm. According to prior approaches, four locations, such as scenes 5, 7, 23, and 30, were eliminated from the training set as a testing set.

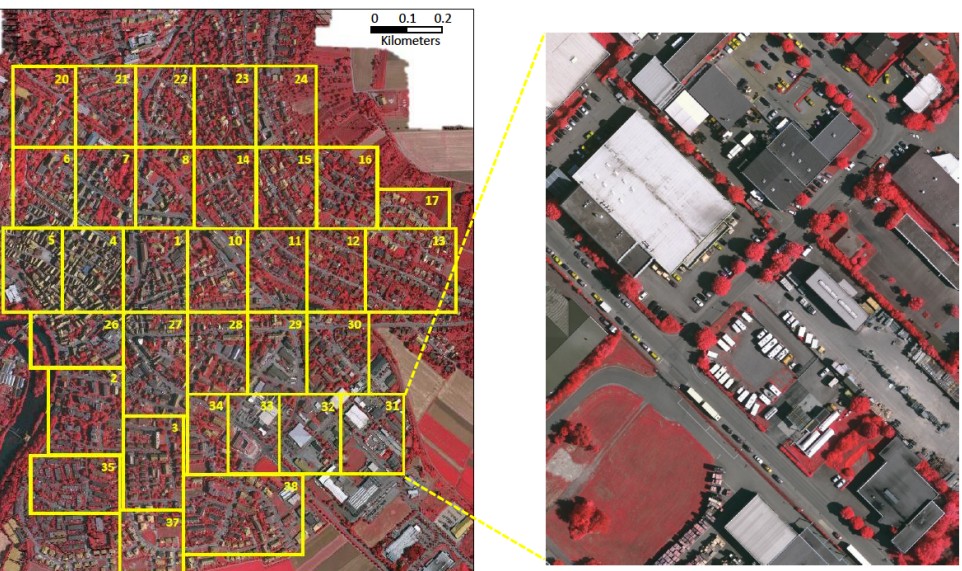

**Figure 5.** Very high-resolution imagery: ISPRS Vaihingen data set.

### 2.2.3. Evaluation Metrics

A true negative (TN) is an outcome where the model predicts the negative class correctly. Similarly, a true positive (TP) is an outcome where the model correctly predicts the positive class. A false negative (FN) is an outcome where the model incorrectly predicts the negative class, and a false positive (FP) is an outcome where the model incorrectly predicts the positive class.

*F*1 is the weighted average of Precision and Recall. Accordingly, this score needs both false negatives and false positives to verify the calculation. However, its *Accuracy* is not straightforward. Although *F*1 is regularly more valuable than *Accuracy*, especially with an uneven class distribution, *Accuracy* is achieved only if false positives and false negatives have similar costs.

It is noted that, for all corpora, the performance of "Pretrained SwinTF with decoder designs" is assessed for *F*1 and *Accuracy*. The *Intersection ovenion* (*IoU*), *F*1, *Precision*, *recall*, and *Accuracy* metrics are used to evaluate class-specific performance; the symphonious average of recall, and accuracy is used to calculate it. The core metrics of *Precision*, *Recall*, *IoU*, *F*1 as well as the *Accuracy*, which divides the number of properly categorized locations by the total number of reference positions are all implemented. Applying Equations (4)–(8), the *Accuracy*, *IoU*, and *F*1 metrics can be expressed as:

$$Accuracy = \frac{TP + TN}{TP + FP + FN + TN} \tag{4}$$

$$Intersection\ over\ Union\ (IoU) = \frac{TP}{TP + FP + FN} \tag{5}$$

$$F1 = \frac{2 \times Precision \times Recall}{Precison + Recall} \tag{6}$$

$$Recall = \frac{TP}{TP + FN} \tag{7}$$

$$Precision = \frac{TP}{TP + FP} \tag{8}$$

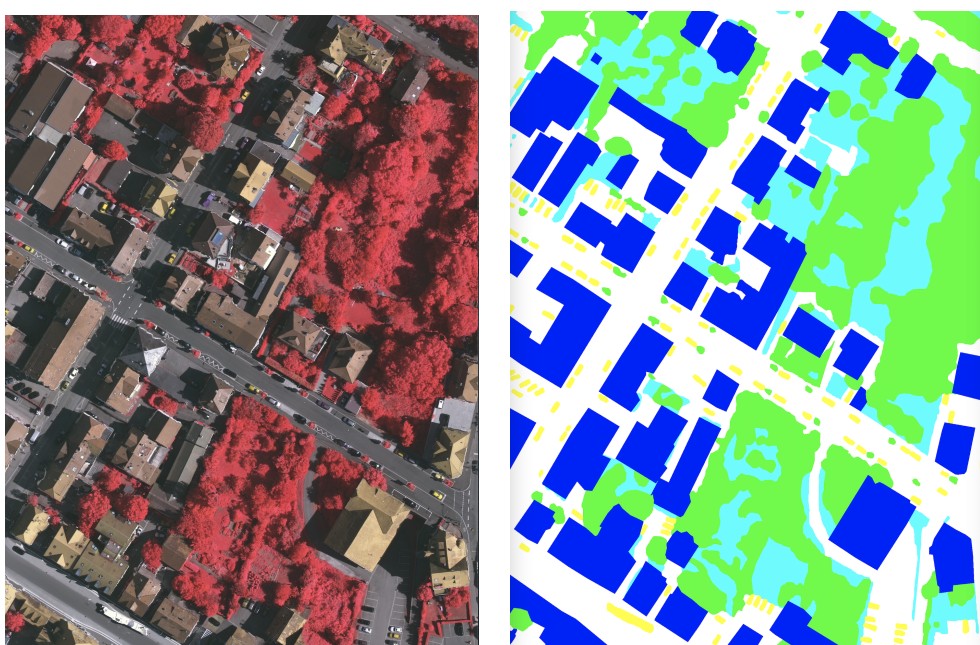

**Figure 6.** This is an example scene from Figure 5. The input image (**left**) depicts an example of an input scene and a target image (**right**). Tree (green), building (blue), jumble/background (red), low vegetation or LV (greenish-blue), and impervious surface or IS (white) are the five categories in the annotated Vaihingen data set.

## 3. Results

Regarding the DL environmental setup, the "TensorFlow Core v2.6.0 (TF)" [42] was created as an end-to-end open-source platform. All experiments were carried out via servers with Intel® Xeon® Scalable 4210R (10 core, 2.4 GHz, 13.75 MB, 100W), 256 GB of memory, and the NVIDIA RTX™ 1080Ti (11 GB) × 2 cards. As designated in Table 2, there are eight procedural acronyms in all proposed designs.

**Table 2.** Acronyms on our proposed scheme strategies.

| Acronym | Representation |
| --- | --- |
| DL | Deep Learning |
| FPN | Feature Pyramid Network |
| LR | Learning Rate |
| PSP | Pyramid Scene Parsing Network |
| ResNet152 | 152-layer ResNet |
| SwinTF | Swin Transformer |
| SwinTF-FPN | Swin Transformer with FPN Decoder Design |
| SwinTF-PSP | Swin Transformer with PSP Decoder Design |
| SwinTF-UNet | Swin Transformer with U-Net Decoder Design |
| TH-Isan Landsat-8 corpus | North East Thailand Landsat-8 data set |
| TH-North Landsat-8 corpus | North Thailand Landsat-8 data set |
| ViT | Vision Transformer |

### 3.1. Results for TH-Isan Landsat-8 Corpus

3.1.1. Effect of Swin Transformer and Pretrained Models

To ensure the contribution of the transformer module, SwinTF was compared with and without Pretrained models on ImageNet-1K by adding or removing the concatenation of this feature in our backbone architecture. The results presented in Tables 3 and 4 suggest that the Pretrained model on ImageNet-1K of Transformer is crucial for the segmentation.

In Table 3, the segmentation *F*1 scores are significantly improved by 3.4% for the backbone networks as compared with SwinTF without Pretrained and GCN-A-FF-DA with Res152.

Furthermore, in Table 4, the impact on the corn is 19.82%; this feature is due to the higher accuracy for almost all classes except the para rubber class. *F*1 scores of 87.74% can still be achieved with the same backbone networks in Table 3 as compared with SwinTF without Pretrained and GCN-A-FF-DA with Res152. Results suggest that the network of the transformer was compatible with end-to-end deep learning.

### 3.1.2. Effect of Transformer with Decoder Designs

To investigate the transformer-based decoder designs, we evaluate our deep architecture with FPN, PSP, and UNet. In Table 3 of our proposed methods, SwinTF-PSP decoder design (the best-proposed model) achieves an *F*1 score of 88.95%, with the FPN decoder design achieving an *F*1 score of 89.80% and with the U-Net decoder design achieving an *F*1 score of 88.30%. Using the same training schedule, our best-proposed model (SwinTF-PSP) significantly outperforms the baselines (GCN), achieving *F*1 of 6.4% and the baselines (Pretrained SwinTF), achieving *F*1 of 2.05% with a clear margin.

Moreover, the decoder designs of our transformers yield concretely better results than original pretrained Swin Transformers. In Table 3 comparing SwinTF-PSP with SwinTF with Pretrained, our best model (PSP decoder designs) achieves 0.14%, 3.85%, and 2.06% improvements for *precision*, *recall*, and *F*1, respectively.

**Table 3.** Results on our testing set: TH-Isan Landsat-8 corpus.

| | Pretrained | Backbone | Model | *Precision* | *Recall* | *F1* | *IoU* |
|---|---|---|---|---|---|---|---|
| Baseline | Yes | - | DeepLab V3 [8] | 0.7547 | 0.7483 | 0.7515 | 0.6019 |
| | Yes | - | UNet [29] | 0.7353 | 0.7340 | 0.7346 | 0.5806 |
| | Yes | - | PSP [30] | 0.7783 | 0.7592 | 0.7686 | 0.6242 |
| | Yes | - | FPN [31] | 0.7633 | 0.7688 | 0.7660 | 0.6208 |
| | Yes | Res152 | GCN-A-FF-DA [36] | 0.7946 | 0.7883 | 0.7909 | 0.6549 |
| | Yes | RestNest-K50-GELU | GCN-A-FF-DA [36,43] | 0.8397 | 0.8285 | 0.8339 | 0.7154 |
| | No | ViT | SwinTF [12,13,37] | 0.8778 | 0.8148 | 0.8430 | 0.7319 |
| | Yes | ViT | SwinTF [12,13,37] | 0.8925 | 0.8637 | 0.8774 | 0.7824 |
| Proposed Method | Yes | ViT | SwinTF-UNet | 0.8746 | 0.8955 | 0.8830 | 0.7936 |
| | Yes | ViT | SwinTF-PSP | 0.8939 | **0.9022** | **0.8980** | **0.8151** |
| | Yes | ViT | SwinTF-FPN | **0.8966** | 0.8842 | 0.8895 | 0.8025 |

**Table 4.** Results on our testing set: TH-Isan Landsat-8 corpus (each class).

| | Pretrained | Backbone | Model | Corn | Pineapple | Para Rubber |
|---|---|---|---|---|---|---|
| Baseline | Yes | - | DeepLab V3 [8] | 0.6334 | 0.8306 | 0.7801 |
| | Yes | - | UNet [29] | 0.6210 | 0.8129 | 0.7927 |
| | Yes | - | PSP [30] | 0.6430 | 0.8170 | 0.8199 |
| | Yes | - | FPN [31] | 0.6571 | 0.8541 | 0.8191 |
| | Yes | Res152 | GCN-A-FF-DA [36] | 0.6834 | 0.8706 | 0.8301 |
| | Yes | RestNest-K50-GELU | GCN-A-FF-DA [36,43] | 0.8982 | 0.9561 | 0.8657 |
| | No | ViT | SwinTF [12,13,37] | 0.7021 | 0.9179 | 0.8859 |
| | Yes | ViT | SwinTF [12,13,37] | 0.9003 | 0.9572 | 0.8763 |
| Proposed Method | Yes | ViT | SwinTF-UNet | 0.9139 | 0.9652 | 0.8876 |
| | Yes | ViT | SwinTF-PSP | **0.9386** | 0.9632 | **0.8985** |
| | Yes | ViT | SwinTF-FPN | 0.9234 | **0.9619** | 0.8886 |

### 3.2. Results for TH-North Landsat-8 Corpus

3.2.1. Effect of Swin Transformer and Pretrained Models

As presented in Tables 5 and 6, the results suggest that the Pretrained model on ImageNet-1K of Transformer proved significant for the segmentation. The results greatly improved the segmentation *F*1 score by 1.06% for the backbone networks and 4.8% for the baseline networks. Furthermore, there was little impact on the para rubber, corn, and pineapple (2%); this feature was due to the higher accuracy for all classes in Table 6. In Table 5, an *F*1 score of 88.73% with the same backbone networks can still be achieved as compared with SwinTF without Pretrained and GCN-A-FF-DA with Res152. This outcome suggests that the network architecture of the transformer was compatible with end-to-end DL.

3.2.2. Effect of Transformer with our Decoder Designs

To examine the transformer-based decoder designs, our deep architecture with FPN, PSP, and UNet was assessed. In Table 5 of our proposed methods, the SwinTF-PSP network (remaining the best-proposed model) achieved an *F*1 score of 63.12%. Further, the FPN decoder design achieved an *F*1 score of 63.06%, and the UNet decoder design achieved an *F*1 score of 62.24%.

**Table 5.** Results on our testing set: TH-North Landsat-8 corpus.

|  | Pretrained | Backbone | Model | *Precision* | *Recall* | *F*1 | *IoU* |
|---|---|---|---|---|---|---|---|
| Baseline | Yes | - | DeepLab V3 [8] | 0.5019 | 0.5323 | 0.5166 | 0.3483 |
|  | Yes | - | UNet [29] | 0.4836 | 0.5334 | 0.5073 | 0.3398 |
|  | Yes | - | PSP [30] | 0.4949 | 0.5456 | 0.5190 | 0.3505 |
|  | Yes | - | FPN [31] | 0.5112 | 0.5273 | 0.5192 | 0.3506 |
|  | Yes | Res152 | GCN-A-FF-DA [36] | 0.5418 | 0.5722 | 0.5559 | 0.3857 |
|  | Yes | RestNest-K50-GELU | GCN-A-FF-DA [36,43] | 0.6029 | 0.5977 | 0.5977 | 0.4289 |
|  | No | ViT | SwinTF [12,13,37] | 0.6076 | 0.5809 | 0.5940 | 0.4225 |
|  | Yes | ViT | SwinTF [12,13,37] | 0.6233 | 0.5883 | 0.6047 | 0.4340 |
| **Proposed Method** | Yes | ViT | SwinTF-UNet | 0.6273 | 0.6177 | 0.6224 | 0.4519 |
|  | Yes | ViT | SwinTF-PSP | **0.6384** | 0.6245 | **0.6312** | **0.4613** |
|  | Yes | ViT | SwinTF-FPN | 0.6324 | **0.6289** | 0.6306 | 0.4606 |

**Table 6.** Results on our testing set: TH-North Landsat-8 corpus (each class).

|  | Pretrained | Backbone | Model | Corn | Pineapple | Para Rubber |
|---|---|---|---|---|---|---|
| Baseline | Yes | - | DeepLab V3 [8] | 0.4369 | 0.8639 | 0.8177 |
|  | Yes | - | UNet [29] | 0.4135 | 0.8418 | 0.7721 |
|  | Yes | - | PSP [30] | 0.4413 | 0.8702 | 0.8032 |
|  | Yes | - | FPN [31] | 0.4470 | 0.8743 | 0.8064 |
|  | Yes | Res152 | GCN-A-FF-DA [36] | 0.4669 | 0.9039 | 0.8177 |
|  | Yes | RestNest-K50-GELU | GCN-A-FF-DA [36,43] | 0.5151 | 0.9394 | 0.8442 |
|  | No | ViT | SwinTF [12,13,37] | 0.5375 | 0.9302 | 0.8628 |
|  | Yes | ViT | SwinTF [12,13,37] | 0.5592 | 0.9527 | 0.8873 |
| **Proposed Method** | Yes | ViT | SwinTF -UNet | 0.5850 | 0.9703 | 0.9117 |
|  | Yes | ViT | SwinTF-PSP | **0.6008** | **0.9877** | **0.9296** |
|  | Yes | ViT | SwinTF-FPN | 0.6006 | 0.9857 | 0.9245 |

Using the same training schedule, our best-proposed model (SwinTF-PSP) significantly outperformed both baselines (GCN), achieving an *F*1 score of 7.52% and the baselines (Pretrained Swin-TF), achieving an *F*1 score of 2.65% by a clear margin. It is evident that the decoder designs of our transformers yielded far better results than the original pretrained

Swin Transformers. In Table 5 comparing SwinTF-PSP with SwinTF with Pretrained, our best model (PSP decoder designs) achieved 1.5%, 3.6%, and 2.6% improvements for *precision*, *recall*, and *F*1, respectively.

### 3.3. Results for ISPRS Vaihingen Corpus

This research aims to take semantic segmentation methods via modern deep learning and apply them to high-resolution geospatial corpora. These differences are summed up in Tables 7 and 8. We noted that our best model (Pretrained SwinTF-FPN) had more robust results on this corpus.

#### 3.3.1. Effect of Swin Transformer and Pretrained Models

In Tables 7 and 8, the results suggest that the Pretrained model on ImageNet-1K of Transformer is also significant for the segmentation. In Table 7, the results much improved the segmentation *F*1-score by 2.73% for the backbone networks and 8.01% for the baseline networks compared with SwinTF without Pretrained and GCN-A-FF-DA with Res152. In Table 8, there was little impact on the impervious surfaces, tree, and car classes at 2.19%, 2.48%, and 9.39%, respectively; this feature was due to the higher accuracy almost of all classes. It is clear that SwinTF-FPN can still achieve *F*1 scores of 94.94% with the same backbone network in Table 7. This result suggests that the network architecture of the transformer was compatible with end-to-end deep learning.

#### 3.3.2. Effect of Transformer with Our Decoder Designs

To investigate the transformer-based decoder designs, our deep architecture was evaluated via FPN, PSP, and UNet, respectively. In Table 7 of our proposed methods, SwinTF-PSP also achieved an *F*1 score of 94.83%. Furthermore, the FPN decoder design (the winner) achieved an *F*1 score of 94.94%, whilst the UNet decoder design achieved an *F*1 score of 94.38%. Using the same training schedule, our best-proposed model (SwinTF-FPN) significantly outperformed both the baselines (GCN), achieving *F*1 of 6.4% and the baselines (Pretrained SwinTF), achieving *F*1 of 2.05% by a clear margin.

**Table 7.** Results on our testing set: ISPRS Vaihingen Corpus.

|  | Pretrained | Backbone | Model | *Precision* | *Recall* | *F*1 | *IoU* |
|---|---|---|---|---|---|---|---|
| Baseline | Yes | - | DeepLab V3 [8] | 0.8672 | 0.8672 | 0.8672 | 0.7656 |
|  | Yes | - | UNet [29] | 0.8472 | 0.8572 | 0.8522 | 0.7425 |
|  | Yes | - | PSP [30] | 0.8614 | 0.8799 | 0.8706 | 0.7708 |
|  | Yes | - | FPN [31] | 0.8701 | 0.8812 | 0.8756 | 0.7787 |
|  | Yes | Res152 | GCN-A-FF-DA [36] | 0.8716 | 0.8685 | 0.8694 | 0.8197 |
|  | Yes | RestNest-K50-GELU | GCN-A-FF-DA [36,43] | 0.9044 | 0.9088 | 0.9063 | 0.8292 |
|  | No | ViT | SwinTF [12,13,37] | 0.8537 | 0.9356 | 0.8770 | 0.7701 |
|  | Yes | ViT | SwinTF [12,13,37] | 0.9756 | 0.8949 | 0.9221 | 0.8753 |
| **Proposed Method** | Yes | ViT | SwinTF-UNet | 0.9203 | 0.9732 | 0.9438 | 0.8977 |
|  | Yes | ViT | SwinTF-PSP | 0.9271 | **0.9820** | **0.9483** | **0.9098** |
|  | Yes | ViT | SwinTF-FPN | **0.9296** | 0.9756 | 0.9494 | 0.9086 |

Moreover, the decoder designs of our transformers yielded much better results than the original pretrained SwinTF. In Table 7, comparing SwinTF-PSP with SwinTF and Pretrained, our best model (FPN decoder design) attained 8.07% and 2.76% improvements for the recall and *F*1, respectively.

Figure 7 shows the prediction results for the entire Isan scene, and Figure 8 shows the prediction results for the entire North scene. Agriculture regions are more dispersed in these zones, and the scenery is more varied. Furthermore, exposed rocks and patches of flora in semiarid environments may have comparable backscatter intensities to structures and be readily misclassified.

**Table 8.** Results on our testing set: ISPRS Vaihingen Corpus (each class).

|  | Model | IS | Buildings | LV | Tree | Car |
|---|---|---|---|---|---|---|
| Baseline | DeepLab V3 [8] | 0.8289 | 0.8026 | 0.8257 | 0.7985 | 0.6735 |
|  | UNet [29] | 0.8189 | 0.7826 | 0.7857 | 0.7845 | 0.6373 |
|  | PSP [30] | 0.8273 | 0.8072 | 0.8059 | 0.8050 | 0.6781 |
|  | FPN [31] | 0.8327 | 0.8111 | 0.8127 | 0.8117 | 0.6896 |
|  | GCN-A-FF-DA [36] | 0.8431 | 0.8336 | 0.8362 | 0.8312 | 0.7014 |
|  | GCN-A-FF-DA [36,43] | 0.9005 | 0.9076 | **0.8942** | 0.8877 | 0.8233 |
|  | SwinTF [12,13,37] | 0.8811 | 0.8934 | 0.8878 | 0.8734 | 0.7866 |
|  | Pretrained SwinTF [12,13,37] | 0.9137 | 0.9139 | 0.8803 | 0.8922 | 0.8118 |
| **Proposed Method** | Pretrained SwinTF-UNet | 0.9139 | 0.9101 | 0.8870 | 0.9035 | 0.9006 |
|  | Pretrained SwinTF-PSP | 0.9259 | **0.9195** | 0.8790 | 0.9093 | 0.9019 |
|  | Pretrained SwinTF-FPN | **0.9356** | 0.9157 | 0.8746 | **0.9169** | **0.9057** |

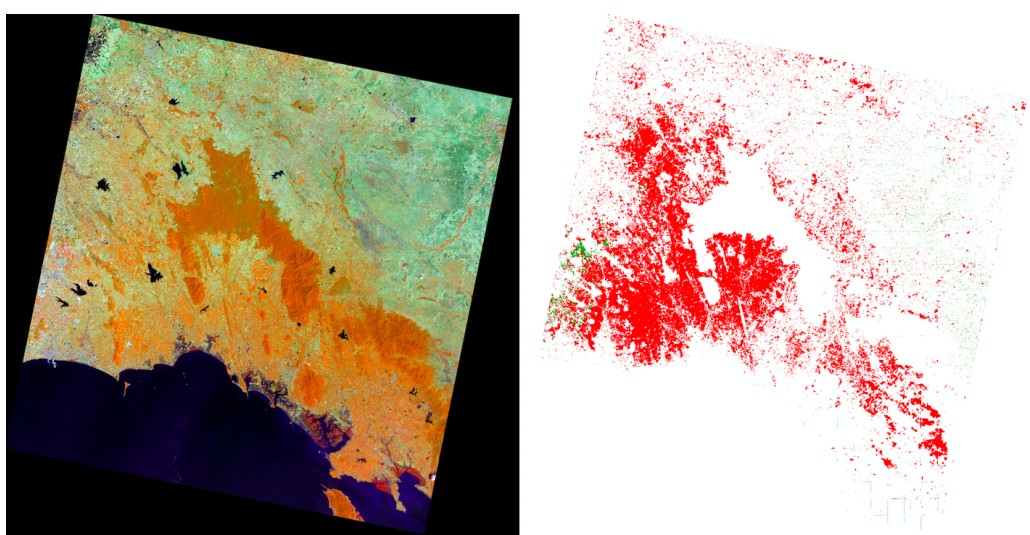

**Figure 7.** Prediction result of "Pretrained SwinTF-PSP" on the entire TH-Isan Landsat-8 corpus scene.

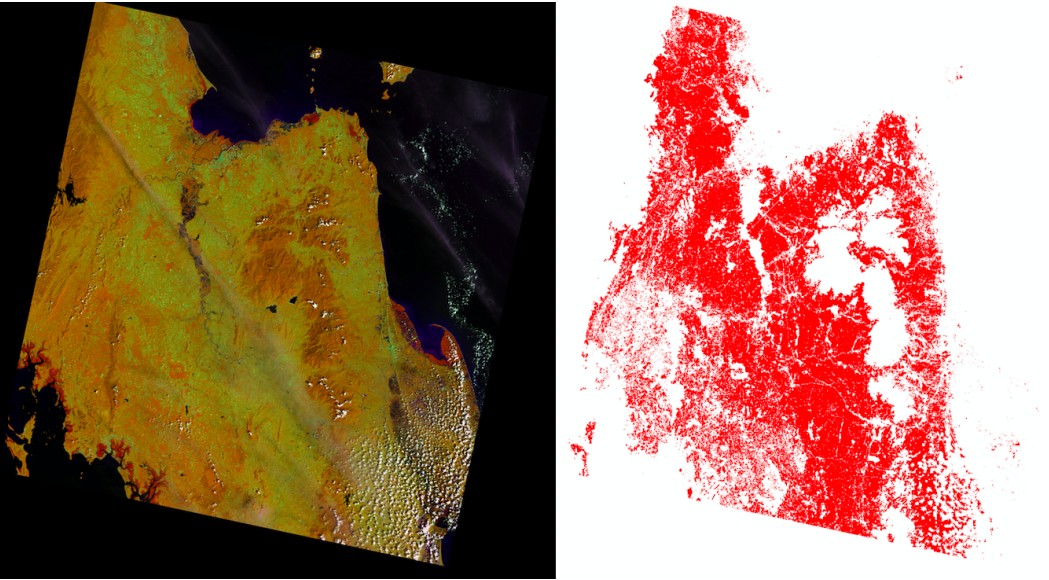

**Figure 8.** Prediction result of "Pretrained SwinTF-PSP" on the entire TH-North Landsat-8 corpus scene.

## 4. Discussion

In this work, the usefulness of SwinTF-based semantic segmentation models for the retrospective reconstruction of Thailand's agriculture region was investigated. For the chosen methodologies, the necessity to prepare sufficient training data may provide certain restrictions.

As shown in Figure 9, the Isan Landsat-8 corpus provided a qualitative segmentation comparison between SwinTF and decoder designs (SwinTF-PSP, SwinTF-FPN, and SwinTF-UNet) and the SOTA baseline (an enhanced GCN). The results of the PSP decoder design demonstrated more precise segmentation for bigger and thinner objects, e.g., the para rubber and corn areas. Moreover, the PSP decoder design also achieved more integrated segmentation on smaller objects, e.g., the pineapple class.

In the validation data of "SwinTF-PSP", Figure 10, there is a more profound disparity (hill) than that in the baseline, Figure 10a. In addition, Figures 10b and 11b show four learning graphs viz. *accuracy*, *precision*, *recall*, and *F*1 lines. The loss line of the "SwinTF-PSP" model appeared deceived (very soft) more than the traditional method in Figure 11a. The number at epoch 99 was selected as a pretrained weight for validation and transfer learning procedures.

As shown in Figure 12, the north Landsat-8 corpus provided a qualitative segmentation comparison between SwinTF and decoder designs (SwinTF-PSP, SwinTF-FPN, and SwinTF-UNet) and the SOTA baseline (an enhanced GCN). The results of the PSP decoder design revealed more precise segmentation for smaller objects, such as the corn area. Moreover, the PSP decoder design also achieved more integrated segmentation on oversized objects, e.g., the para rubber class.

There is a more profound disparity (hill) in the validation data of "SwinTF-PSP", Figure 13, than that in the baseline, Figure 13a. In addition, Figures 13b and 14b depict the four learning lines viz. *accuracy*, *precision*, *recall*, and *F*1 lines. The loss line of the "SwinTF-PSP" model appeared deceived (very soft) more than the traditional method in Figure 14a. The number at epoch 100 was chosen as a pretrained weight for validation and transfer learning procedures.

As shown in Figure 15, the ISPRS Vaihingen corpus provided qualitative segmentation comparison between SwinTF and decoder designs (SwinTF-PSP, SwinTF-FPN, and SwinTF-UNet) and the SOTA baseline (an enhanced GCN). The results of the FPN decoder design exhibited more precise segmentation for smaller objects, e.g., the car and tree (classes). Moreover, the PSP decoder design achieved more integrated segmentation on oversized objects, e.g., impervious surfaces. The number at epoch 99 was picked as a pretrained weight for validation and transfer learning procedures.

In Figure 16, there was a more profound disparity (hill) in the validation data of "SwinTF-PSP" than that in the baseline, Figure 16a. In addition, Figures 16b and 17b show the four learning lines viz. *accuracy*, *precision*, *recall*, and *F*1 lines. The loss line of the "SwinTF-PSP" model appeared deceived (very soft) more than the traditional method in Figure 17a. The number at epoch 95 was picked as a pretrained weight for validation and transfer learning procedures.

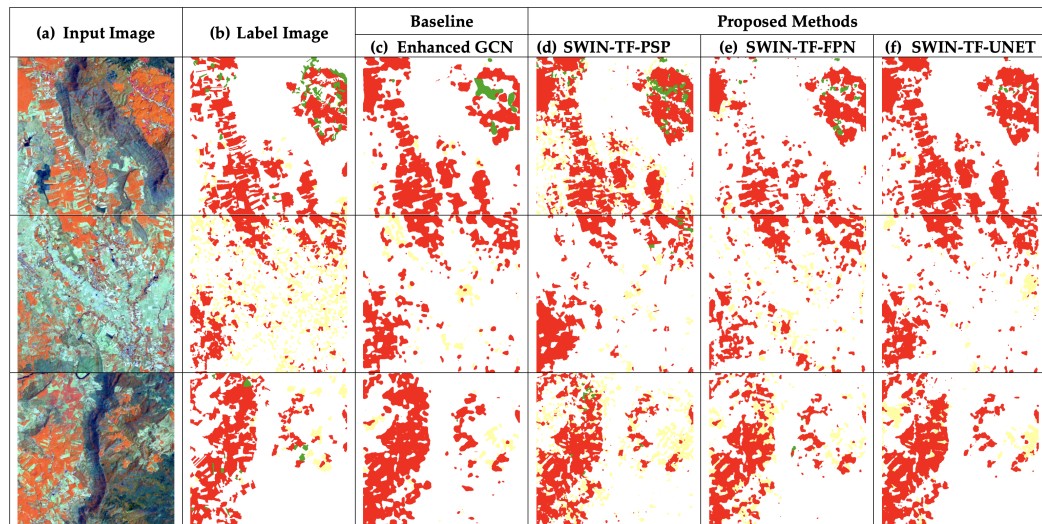

**Figure 9.** Comparisons between our proposed methods and baseline for the TH-Isan Landsat-8 corpus testing set.

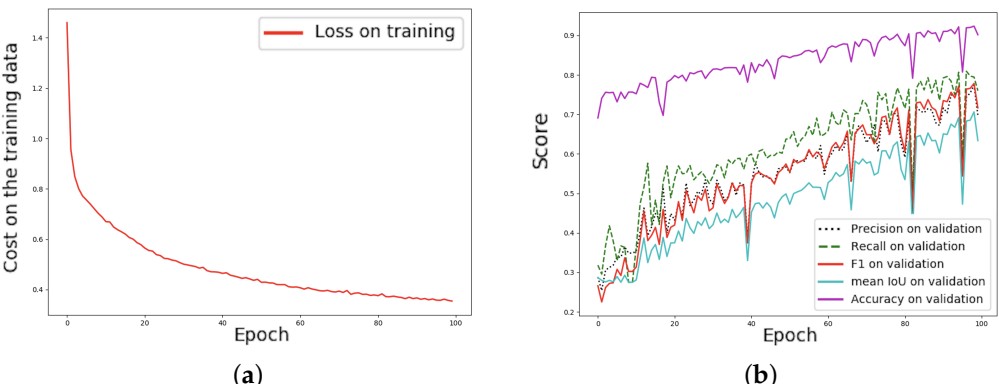

**Figure 10.** Graph (learning curves): on TH-Isan Landsat-8, the proposed approach, and SwinTF-PSP (**a**) Plot of model loss (cross-entropy) on training and testing corpora; (**b**) performance plot on the testing corpus.

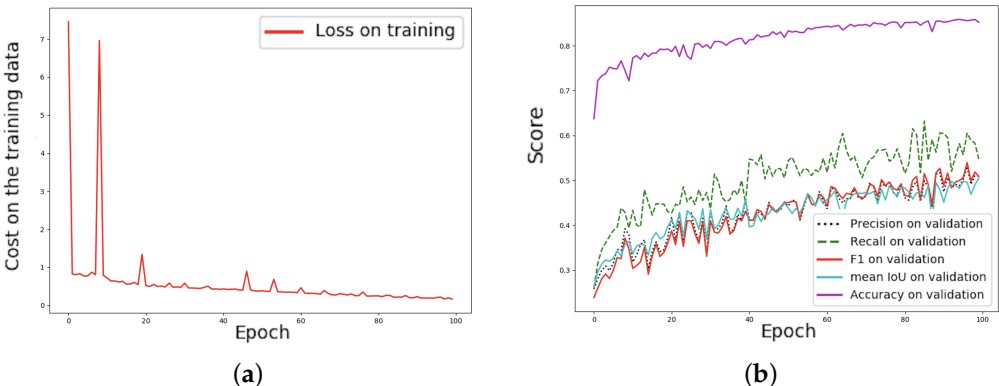

**Figure 11.** Graph (learning curves): TH-Isan Landsat-8 corpus, the baseline approach, and SwinTF (**a**) Plot of model loss (cross-entropy) on training and testing corpora; (**b**) performance plot on the testing corpus.

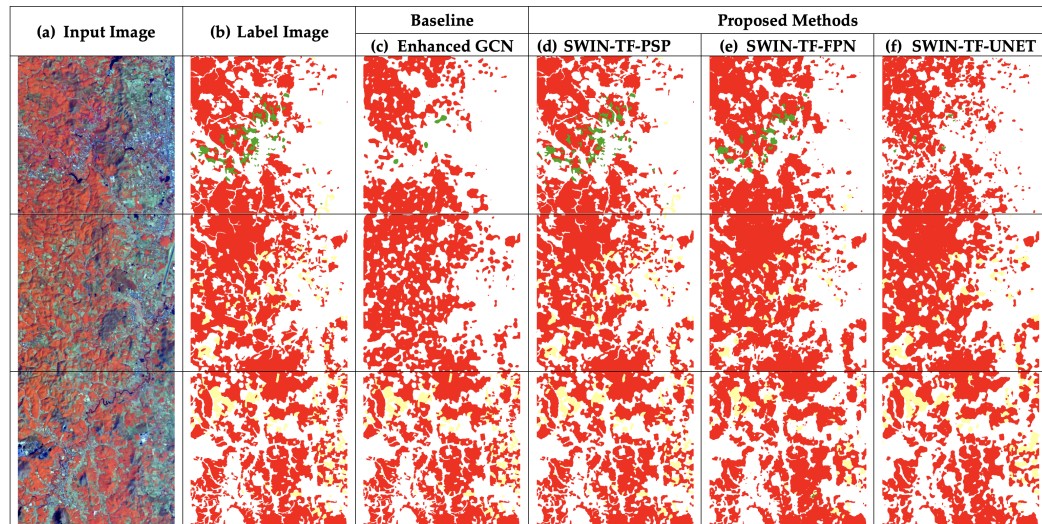

**Figure 12.** Comparisons between our proposed methods and baseline for the TH-North Landsat-8 corpus testing set.

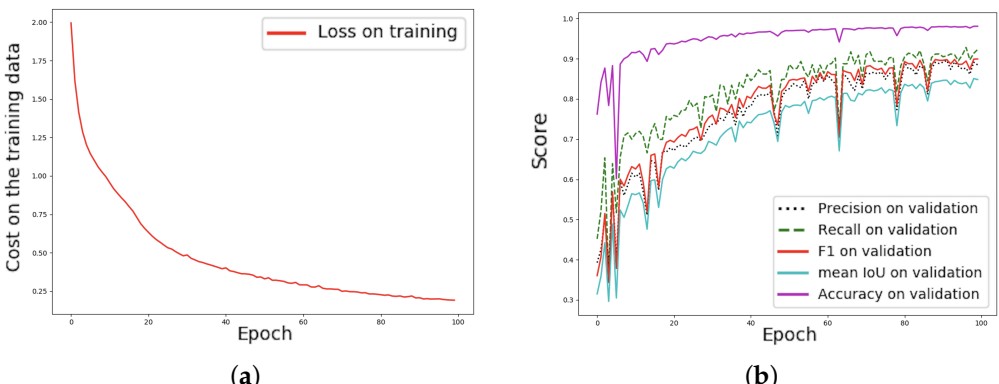

(**a**)                                                                                     (**b**)

**Figure 13.** Graph (learning curves): TH-North Landsat-8 corpus, the proposed approach, and SwinTF-PSP (**a**) Plot of model loss (cross-entropy) on training and testing corpora; (**b**) performance plot on the testing corpus.

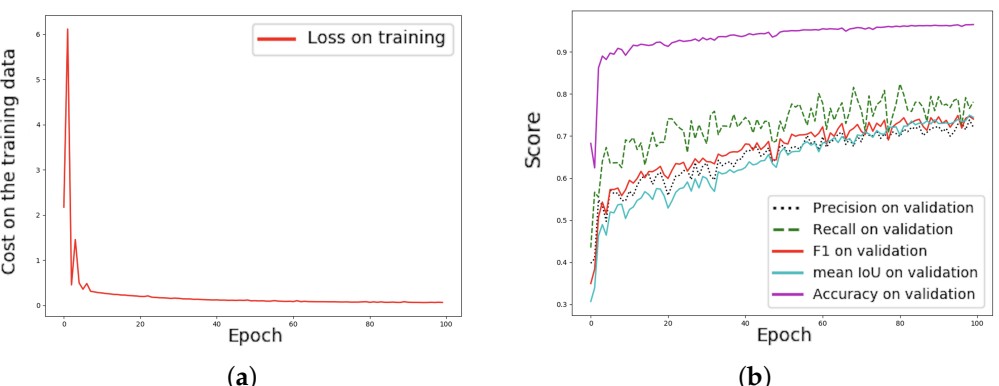

(**a**)                                                                                     (**b**)

**Figure 14.** Graph (learning curves): TH-North Landsat-8 corpus, the baseline approach, and SwinTF (**a**) Plot of model loss (cross-entropy) on training and testing corpora; (**b**) performance plot on the testing corpus.

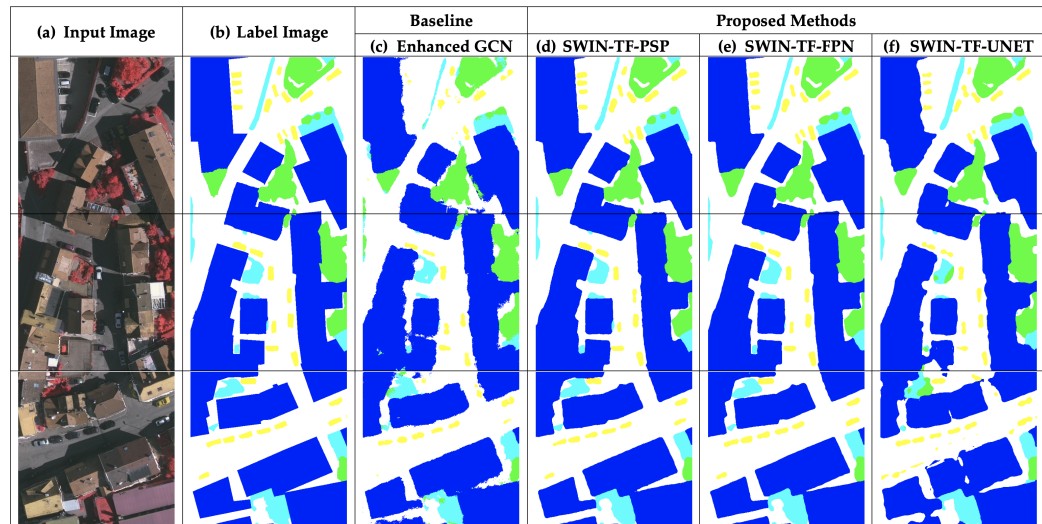

**Figure 15.** Comparisons between our proposed methods and baseline for the ISPRS Vaihingen corpus testing set.

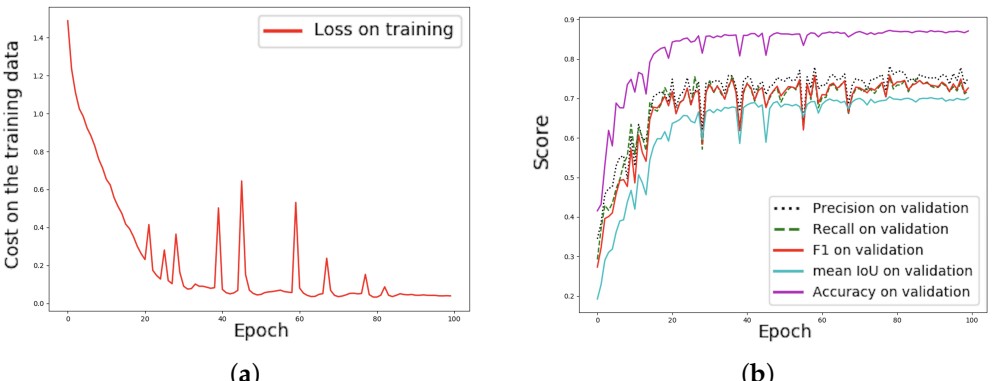

**Figure 16.** Graph (learning curves): ISPRS Vaihingen corpus, the proposed approach, and SwinTF-FPN (**a**) Plot of model loss (cross-entropy) on training and testing corpora; (**b**) performance plot on the testing corpus.

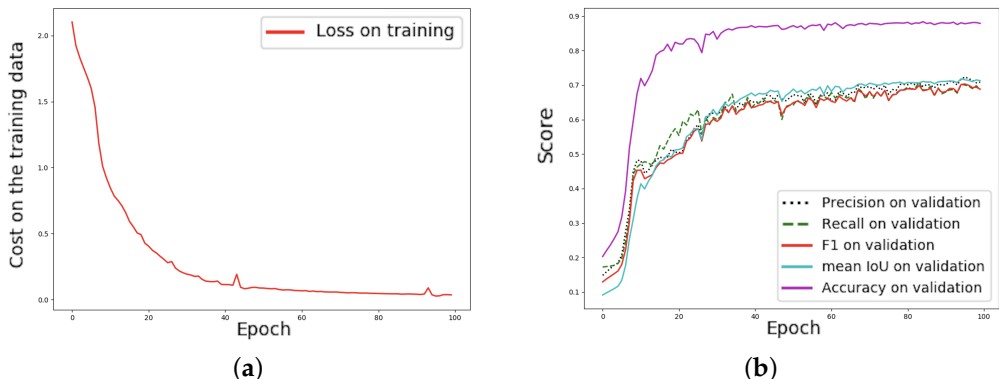

**Figure 17.** Graph (learning curves): ISPRS Vaihingen corpus, the baseline approach, and SwinTF (**a**) Plot of model loss (cross-entropy) on training and testing corpora; (**b**) performance plot on the testing corpus.

*Limitations and Outlook*

In this research, the appropriateness of transformer-based semantic segmentation models for the retrospective reconstruction of cultivation (corn, pineapple, and para rubber) in Thailand as well as the ISPRS Vaihingen data set (aerial images) was investigated. For

the selected methods, the requirement to prepare extensive training data may pose some limitations. For future studies, achieving high performance with limited training data using our approach must be cost-effective for multi-temporal agriculture mapping.

Therefore, further evaluation of the effectiveness of using modern DL methods with Landsat-8 data beyond a national scale is required. Notwithstanding some limitations, this study adds a baseline, including DeepLab v3, PSP, FPN, and UNet, for proving our best model performance. In future work, more varieties of modern image labeling, as well as some analytical perspectives, e.g., evolving reinforcement learning (RL) algorithms, generative adversarial networks (GANs), or quantization methods for efficient neural network inference, will be reviewed and analyzed for such tasks.

## 5. Conclusions

This paper exhibits an alternative viewpoint for semantic segmentation by prefacing decoder designs for transformer models. The experimental results show that (1) the pretrained transformer models on ImageNet-1K achieved good results for both the Landsat-8 (medium resolution) and ISPRS Vaihingen corpus (very high-resolution). The $F$1-scores were found to range from 84.3% to 87.74%, 59.4% to 64.47%, and 87.7% to 92.21% for the Isan, Nan, and ISPRS Vaihingen corpora, respectively. (2) Our results were compared with other decoder design methods, including FPN, PSP, and U-Net.

It is evident that the proposed approach proved its worth as a dependable technique. Our detailed qualitative and quantitative investigations on three complex remote sensing tasks revealed that both FPN and PSP decoder designs consistently outperformed the baselines and state-of-the-art techniques, thus, demonstrating their significant efficacy and capabilities. In addition, the average accuracy was better than 90% for almost all classes of the data sets.

**Author Contributions:** Conceptualization, T.P.; Formal analysis, T.P.; Investigation, T.P.; Methodology, T.P.; Project administration, T.P.; Resources, T.P.; Software, T.P.; Supervision, T.P. and P.V.; Validation, T.P., K.J., S.L. and P.S.; Visualization, T.P.; Writing—original draft, T.P.; Writing—review and editing, T.P. and P.V. All authors have read and agreed to the published version of the manuscript.

**Funding:** This research was supported by the Ratchadapisek Somphot Fund for Postdoctoral Fellowship, Chulalongkorn University.

**Acknowledgments:** Teerapong Panboonyuen, also known as Kao Panboonyuen appreciates (thanks) and acknowledges the scholarship from Ratchadapisek Somphot Fund for Postdoctoral Fellowship, Chulalongkorn University, Thailand.

**Conflicts of Interest:** The authors declare no conflict of interest.

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
