# Peer review of "Transformer-Based Decoder Designs for Semantic Segmentation on Remotely Sensed Images"

_remotesensing, doi:10.3390/rs13245100_

Round 1
Reviewer 1 Report
Could you please release the source code of the paper on the GitHub link as you describe in your abstract?
Author Response
We appreciate your detailed comments and suggestions. There are identify some important points which we hope to clarify and address here and in our revision via this Google Drive link.
https://docs.google.com/document/d/1UV0Thes4gueafpcWM5oUxH9WZFbE0qd8TVDgfEBcnQg/edit?usp=sharing
Q) Could you please release the source code of the paper on the GitHub link as you describe in your abstract?
A) Our source code includes Landsat-8 data sets licensed under the Thai space agency and space research organization, The Geo-Informatics and Space Technology Development Agency (Public Organization). So, it will be provided by request with their approval. However, as we describe in the abstract, our GitHub link (only the source code of our SwinTF with decoder designs) will immediately provide total codes after publishing our latest article.

Reviewer 2 Report
This paper presents an alternative viewpoint for image labeling by using pre-existing decoder designs with the pre-existing Transformer module (SwinTF). Even though this paper presents an interesting and useful transformer-based decoder design for semantic segmentation, there seem to be some major problems in terms of organization and explanations of different concepts. In my opinion, this paper in its current form is not adequate to be published and should be resubmitted after investing additional effort to strengthen some issues pointed below.
Please consider answering some of the following issues.
In the paper, you write that ViT complexity is quadratic to the size of an image, while your solution is linear and operated regionally. Can you elaborate more on this? Why is ViT's complexity quadratic, and how is your linear? How does this help to model correlation in visual signals?
Section 3. describes the proposed methodology. Even though the idea is interesting, the entire section is not written well, and it is unnecessarily challenging to follow. The explanations of the methodology are not described well and the connection with figures is not existing. Please make an effort to make this section more readable. It is important to make clear distinguishes between your work and SwinTF.
There is no difference between left and right images in Figure 1., except input and out images? If this is true, there is no need to have two architectures. In this case, figure 2. with additions of two different I/O might be sufficient. Please consider removing Figure 1., and improving Figure 2.
In subsection 3.2 it is not clear if you have modified the original decoder designs and to what degree? All modifications should be denoted in Figure 2 (to some degree at least). Illustrations of decoders are very basic.
In subsection 4.1, the connection between 1,420 satellite images and 8,785 + 785 images is not clear. What do the three classes represent? Please elaborate.
You perform experiments on three datasets: one public and two private. You provide comparison results for two private datasets against ref [26] (your previous work) and [41] (called baseline in the paper), in tables 3, 4, 5, and 6. The difference between your current, previous, and [41] work is not clear. You do not provide the same comparison in the case of ISPRS Vaihingen Corpus, which is the public dataset and these results are more important than two private datasets. Tables 7 and 8 present results for the TH-Isan Landsat-8 corpus? This is confusing. Please make an effort to strengthen this part and clearly present why your work is better than state of the art.
A list of some mistakes is listed below. Please note that this is not the full list of mistakes and errors noticed in your paper. Please make an effort to find and correct all mistakes and errors, and make this paper more readable.
Grammar mistakes and errors:
In line 1, "dwells" should be replaced with is.
In line 22, "anxieties" should be replaced with the more adequate word.
In line 34, A "." should be deleted.
In line 37, "It matures" is unusual wording. The entire sentence is not formulated well. How are receptive fields limited?
In line 42, "that's" is not adequate. The entire sentence is not formulated well. What does it mean "in the first area"?
In line 43, "it has matured famous"? The entire paragraph should be rewritten.
In line 48, you write "In terms of real-world precision, this method is also effective." What does real-world precision mean? How is it effective?
In line 50, change "segmenting multi-objects" to "multi-object segmentation".
In line 64, What does it mean when you say that "SwinTF with decoder designs overcomes GCN based architectures"? This should be reformulated.
In line 75, "its model is inappropriate for profit as a general-purpose backbone..." is not formulated well. For profit?
In line 89, how does your SwinTF "attends"?
In line 91, what do you mean by "mortal language"? The entire sentence is not formulated well.
In line 92, what kind of mismatch exists between 1D seq. and 2D image?
In line 96, this sentence does not make any sense.
In line 97, what global figure?
In line 100, this sentence does not make any sense. The entire paragraph is not formulated well.
In line 107, "Furnished the quadratic network complexity of the ViT..." does not make any sense.
In line 109, "Accordingly, tokenizing every pixel of the image as input to our SwinTF is out of the puzzle." Out of the puzzle?
In line 112, there is an error in the formula? When compared with 114.
Author Response
We appreciate your detailed comments and suggestions. There are identify some important points which we hope to clarify and address here and in our revision via this Google Drive link.
https://docs.google.com/document/d/1UV0Thes4gueafpcWM5oUxH9WZFbE0qd8TVDgfEBcnQg/edit?usp=sharing
Q1) In the paper, you write that ViT complexity is quadratic to the size of an image, while your solution is linear and operated regionally. Can you elaborate more on this? Why is ViT's complexity quadratic, and how is your linear? How does this help to model correlation in visual signals?
A1) We sincerely appreciate your additional feedback. We have revised this sentence to “it usually takes high computational costs for the previous transformer network e.g. Pyramid ViT, which is quadratic to the size of an image. In contrast, SwinTF has solved the computational issue and costs only linear to the image size. Also, SwinTF has improved the accuracy by operating the model regionally, enhancing receptive fields that highly correlate to visual signals.”
It means the previous network (for example, Pyramid Vision Transformer) is still quadratic to the size of an image, but the SwinTF has already fixed this issue. It helps to model correlation in visual signals because images are in much higher resolution, and vision tasks that require pixel-level predictions are intractable for transformers.
** Wang, Wenhai, et al. "Pyramid vision transformer: A versatile backbone for dense prediction without convolutions." arXiv preprint arXiv:2102.12122 (2021).
** Liu, Ze, et al. "Swin transformer: Hierarchical vision transformer using shifted windows." arXiv preprint arXiv:2103.14030 (2021).
Q2) Section 3. describes the proposed methodology. Even though the idea is interesting, the entire section is not written well, and it is unnecessarily challenging to follow. The explanations of the methodology are not described well and the connection with figures is not existing. Please make an effort to make this section more readable. It is important to make clear distinguishes between your work and SwinTF.
A2) We are happy to hear that you found our SwinTF comprehensive and convincing and that you agree that our insights are interesting. You brought up some great questions and suggestions that we discuss further below. Please let us know if further clarification is required. We appreciate your detailed comments and suggestions. They identify some important points especially The explanations of the methodology and the connection with figures of our SwinTF which we hope to clarify and address here and in our revision. We have already redrawn our proposed method in both Figure 1 (Overall architecture of our Swin Transformer (SwinTF).) and Figure 2 (Overall architecture of our SwinTF-UNet, SwinTF-PSP, and SwinTF-FPN).
Q3) There is no difference between left and right images in Figure 1., except input and out images? If this is true, there is no need to have two architectures. In this case, figure 2. with additions of two different I/O might be sufficient. Please consider removing Figure 1., and improving Figure 2.
We totally agree with your comments. Yes, there is no need to have two architectures. And, we have already removed the right network in Figure 1 with described the whole of our SwinTF only one diagram. Also, we have already improved Figure 2.
A3) We also improved the figure layout of the paper in the revised manuscript (see revised manuscript attached, changes marked in yellow).
Q4) In subsection 3.2 it is not clear if you have modified the original decoder designs and to what degree? All modifications should be denoted in Figure 2 (to some degree at least). Illustrations of decoders are very basic.
A4) Thank you very much for your suggestion. Since the performance of SwinTF is limited due to its decoder, we aim to improve the performance by applying and comparing several modern decoder networks (our SwinTF-UNet, SwinTF-PSP, and SwinTF-FPN). Also, Figure 2 has been revised to make readers better understand our networks.
Q5) In subsection 4.1, the connection between 1,420 satellite images and 8,785 + 785 images is not clear. What do the three classes represent? Please elaborate.
A5)Thank you very much for your helpful feedback. Sorry for any indistinction. As mentioned in the general rebuttal, we agree with this unclear number, e.g., 1,420. So, We have created and added Table 1 (Numbers of training, validation, and testing sets) in Subsection 2.2 Aerial and Satellite Imagery for representing all of the numbers that appear in our article. Also, three classes represent corn, pineapple, and para rubber (described as Figure 3 and Subsection 2.2.1 North East (Isan) and North of Thailand Landsat-8 Corpora).
Q6) You perform experiments on three datasets: one public and two private. You provide comparison results for two private datasets against ref [26] (your previous work) and [41] (called baseline in the paper), in tables 3, 4, 5, and 6. The difference between your current, previous, and [41] work is not clear. You do not provide the same comparison in the case of ISPRS Vaihingen Corpus, which is the public dataset and these results are more important than two private datasets. Tables 7 and 8 present results for the TH-Isan Landsat-8 corpus? This is confusing. Please make an effort to strengthen this part and clearly present why your work is better than state of the art.
A6) We are deeply sorry for our mistake. Actually, we had a comparison in the case of ISPRS Vaihingen Corpus in Table 7 and Table 8, but it is the mistake of our table name and also we have already revised the table name.

Reviewer 3 Report
The paper addresses a transformer-based decoder for semantic segmentation on remote sensing images. The paper is well-written and presents interesting results. I would suggest the following revisions for its publication.
The authors could make a single table with all acronyms and abbreviations. It will make it easier for the readers. Besides, the authors could also provide an abbreviation to deep learning as DL since it appears with much frequency in the text. I suggest the authors take a last look at all words that could use abbreviations.
The expression “image labeling” confuses the readers. Image labeling is often referred to as the process of making the ground truth data. There are 11 recurrences of this term in the paper. Would you please check all examples and choose another term when referring to the process of generating pixel-wise predictions?
In the Evaluation metrics section (line 227), the authors could briefly introduce the confusion matrix. Besides, there is no need to say “false positive (abbreviated as FP).” The authors could write “false positive (FP).” Also, in line 237, it is written “… equations (7-4)” I believe the authors meant 4-7.
The Intersection over Union (IoU) metric is one of the most used for the semantic segmentation task and would be interesting to include in the results.
The model configurations (learning rate, batch size, etc.) is missing. What loss function was used? The authors mention the number of epochs in the experiments section. Those parameters should be explained in the material and methods section.
The authors should inform the readers of the number of images used for training and validation in the ISPRS Vaihingen Corpus dataset. Besides, why didn’t the authors use a test set?
The authors use different architectures using the transformers (e.g., Unet, PSPNet, FPN). I strongly suggest the authors compare other semantic segmentation models as well. For example, the authors evaluated the GCN-A-FF-DA. Why not compare with the Unet, FPN, PSPNet, DeepLabv3+? Since the authors are proposing a new method, seeing the differences with more models would be very interesting.
This journal adopts the following structure: introduction, materials and methods, results, discussion, and conclusion (https://www.mdpi.com/journal/remotesensing/instructions). Sections 4 and 5 should be in the materials and methods section. Usually, it is more common to talk about the image before talking about the deep learning model. However, using the exact flow of the authors, I would suggest the following division for the Materials and methods section:
3. Material and Methods
3.1 Transformer Model
3.1.1 Transformer Based Semantic Segmentation
3.1.2 Decoder Designs
3.1.3 Environment and Deep learning configurations
The authors could include here lines 240-244 and the model hyperparameters, which is missing.
3.2 Aerial and Satellite Imagery
3.2.1 North East (Isan) and North of Thailand Landsat-8 Corpora
3.2.2 ISPRS Vaihingen Corpus
3.3 Evaluation Metrics
The experiments section should be divided into results and discussion. Besides, it is crucial to compare the current work and other related works. Even though there are tables, the authors should write interpretations on the discussion. How is this method better than others? How does it compare to other methods? How does this work make advancements in the remote sensing field? The limitations could also be a subsection in the discussion.
Author Response
We appreciate your detailed comments and suggestions. There are identify some important points which we hope to clarify and address here and in our revision via this Google Drive link.
https://docs.google.com/document/d/1UV0Thes4gueafpcWM5oUxH9WZFbE0qd8TVDgfEBcnQg/edit?usp=sharing
Q1) The authors could make a single table with all acronyms and abbreviations. It will make it easier for the readers. Besides, the authors could also provide an abbreviation to deep learning as DL since it appears with much frequency in the text. I suggest the authors take a last look at all words that could use abbreviations.
A1) We appreciate your detailed comments and suggestions. There are identified some essential points which we hope to clarify and address here and in our revision. Also, we have revised and removed duplicated abbreviations/acronyms tables into only a single table with; remaining only Table 2 (Acronyms on our proposed scheme strategies) in Section 3.
Q2) The expression “image labeling” confuses the readers. Image labeling is often referred to as the process of making the ground truth data. There are 11 recurrences of this term in the paper. Would you please check all examples and choose another term when referring to the process of generating pixel-wise predictions?
A2) Yes, we have already checked. To be understandable to the reader, we have changed the phrases ‘image labeling’ to ‘semantic segmentation’ that means classifying each pixel belonging to a particular label.
Q3) In the Evaluation metrics section (line 227), the authors could briefly introduce the confusion matrix. Besides, there is no need to say “false positive (abbreviated as FP).” The authors could write “false positive (FP).” Also, in line 237, it is written “… equations (7-4)” I believe the authors meant 4-7.
A3) We totally agree with your comments. We have already revised the whole paragraph to “A true negative (TN) is an outcome where the model predicts the negative class correctly. Similarly, a true positive (TP) is an outcome where the model correctly predicts the positive class. A false negative (FN) is an outcome where the model incorrectly predicts the negative class, and a false positive (FP) is an outcome where the model incorrectly predicts the positive class.” instead, to make it easy for the reader to understand.
Q4) The Intersection over Union (IoU) metric is one of the most used for the semantic segmentation task and would be interesting to include in the results.
A4) We have already added the Intersection over Union (IoU) score into all Tables of our experiments.
Q5) The model configurations (learning rate, batch size, etc.) is missing. What loss function was used? The authors mention the number of epochs in the experiments section. Those parameters should be explained in the material and methods section.
A5) We agree with this reviewer to explain all our hyperparameters in the material and methods section. So, we have already written and added subsection 2.3.1 Environment and Deep learning Configurations for the whole model configurations of our SwinTF.
Q6) The authors should inform the readers of the number of images used for training and validation in the ISPRS Vaihingen Corpus dataset. Besides, why didn’t the authors use a test set?
A6) Sorry for any indistinction. As mentioned in Subsection 2.2.2 ISPRS Vaihingen Corpus, it means the testing set, but we noticed that the remote sensing field often used the phrased "a validation set" instead of "the test set." However, to be clear, we have already fixed this phrase to the testing set. So, once again: Sorry and thanks!
Q7) The authors use different architectures using the transformers (e.g., Unet, PSPNet, FPN). I strongly suggest the authors compare other semantic segmentation models as well. For example, the authors evaluated the GCN-A-FF-DA. Why not compare with the Unet, FPN, PSPNet, DeepLabv3+? Since the authors are proposing a new method, seeing the differences with more models would be very interesting.
A7) Thank you for your suggestions about comparing these baselines (UNet, FPN, PSPNet, DeepLabv3+). So, we have a new experiment and have already added UNet, FPN, PSPNet, DeepLabv3+ into our baselines in Table 3,4,5,6,7, and 8. And, we agreed that seeing the differences with more models would be very interesting.

Reviewer 4 Report
In this paper, the authors have proposed a transformer-based decoder design for semantic segmentation of remotely sensed images. Overall, the paper is well written but requires some major improvements.
1) Novelty of the work is marginal. Please try to improve it and expand the contributions section at least to four-folds.
2) Please include a detailed comparison table that should highlight the pros and cons of the previous methods as well as the proposed method. Also, include a description column in the table that should give the core idea of each method.
3) Summarise the details related to the datasets in the table that should include the names of all the datasets used. It should mention the total number of images, training and testing images, etc.
4) Please improve the quality of the figures specifically text in the figures (7,8,10,11,13,14) is very hard to read.
5) Please mention the reference numbers of the methods and datasets when you are mentioning them in figures and tables.
6) In comparison diagrams, please mention the colors used to represent the TP, FN etc.
Author Response
We appreciate your detailed comments and suggestions. There are identify some important points which we hope to clarify and address here and in our revision via this Google Drive link.
https://docs.google.com/document/d/1UV0Thes4gueafpcWM5oUxH9WZFbE0qd8TVDgfEBcnQg/edit?usp=sharing
Q) In this paper, the authors have proposed a transformer-based decoder design for semantic segmentation of remotely sensed images. Overall, the paper is well written but requires some major improvements.
A) We appreciate your detailed comments and suggestions. They identify some important points which we hope to clarify and address here and in our revision. We also improved the figure layout of the paper in the revised manuscript (see revised manuscript attached, changes marked in yellow).
Q1) Novelty of the work is marginal. Please try to improve it and expand the contributions section at least to four-folds.
A1) We totally agree. And, we have expanded and rearranged the contributions section six-fold at Subsection 2 Material and Methods.
Q2) Please include a detailed comparison table that should highlight the pros and cons of the previous methods as well as the proposed method. Also, include a description column in the table that should give the core idea of each method.
A2) We have included a detailed comparison table in Subsection 4 Discussion. And we have rewritten the core idea of each method into Subsection 1 Introduction (Bullet Part), and Also, we have written a new Subsection 4.1 Limitations and Outlook to highlight the pros and cons of the previous methods as well as the proposed method.
Q3) Summarise the details related to the datasets in the table that should include the names of all the datasets used. It should mention the total number of images, training and testing images, etc.
A3) You are totally correct about the use of the table. We have already created and added Table 1 (Numbers of training, validation, and testing sets) in Subsection 2.2 Aerial and Satellite Imagery for representing all of the numbers that appear in our article include the names of all the datasets the total number of images, training, and testing images.
Q4) Please improve the quality of the figures specifically text in the figures (7,8,10,11,13,14) is very hard to read.
We totally agree with your comments on the figures, specifically the text in the figures (7,8,10,11,13,14).
A4) We have already redrawn improved the quality of the figures, specifically the text in the figures (7,8,10,11,13,14). Also, it is now effortless to read.
Q5) Please mention the reference numbers of the methods and datasets when you are mentioning them in figures and tables.
A5) We have made it more explicit about rewriting the reference numbers of the methods and datasets when we mention them in figures and tables in Subsection 3 Results including 3.1.1 Effect of Pretrained of Swin Transformer on The Results on TH-Isan Landsat-8 Corpus, 3.1.2 Effect of Transformer with Decoder Designs on The Results on TH-Isan Landsat-8 Corpus, 3.1.3 Effect of Pretrained of Swin Transformer on The Results on TH-North Landsat-8 Corpus, 3.1.4 Effect of Transformer with Decoder Designs on The Results on TH-North Landsat-8 Corpus, 3.1.5 Effect of Pretrained of Swin Transformer on The Results on ISPRS Vaihingen Corpus, 3.1.6 Effect of Transformer with Decoder Designs on The Results on ISPRS Vaihingen Corpus.
Q6) In comparison diagrams, please mention the colors used to represent the TP, FN etc.
A6) We have made the color in comparison diagrams more explicit, meaning to represent each class, e.g., corn, para-rubber, and pineapple class. Nevertheless, it could compare the color of our prediction with the ground-truth image in column (b) Label Image without considering the colors used to represent the TP, FN.

Round 2
Reviewer 3 Report
The authors attended to all of my demands and the paper is suitable for publication.
Reviewer 4 Report
Most of my comments are addressed. I recommend the acceptance of the paper in present form.